# Advances in Research on the Relationship between Vaginal Microbiota and Adverse Pregnancy Outcomes and Gynecological Diseases

**DOI:** 10.3390/microorganisms11040991

**Published:** 2023-04-11

**Authors:** Fuju Zhao, Xianyang Hu, Chunmei Ying

**Affiliations:** 1Clinical Laboratory, Huadong Hospital, Fudan University, Shanghai 200031, China; 2Clinical Laboratory, Obstetrics and Gynecology Hospital, Fudan University, Shanghai 200011, China; 3Huadong Hospital, Fudan University, Shanghai 200031, China

**Keywords:** vaginal microbiota, pregnancy outcome, reproductive health

## Abstract

The human microbiota inhabiting different parts of the body has been shown to have a significant impact on human health, with the gut microbiota being the most extensively studied in relation to disease. However, the vaginal microbiota is also an essential commensal microbiota in the female body that plays a crucial role in female health. Despite receiving less attention than gut microbiota, its importance in regulating reproductive immunity and its complex dynamic properties have been increasingly recognized in recent years. Advances in research on the relationship between vaginal microbiota and pregnancy outcomes & gynecological diseases in women have shed light on the importance of maintaining a healthy vaginal microbiota. In this review, we aim to compile recent developments in the study of the vaginal microbial ecosystem and its role in female health and reproductive outcomes. We provide a comprehensive account of the normal vaginal microbiota, the association between the vaginal microbiota and pregnancy outcomes, and the impact of the vaginal microbiota on gynecological diseases in women. By reviewing recent research, we hope to contribute to the advancement of academic medicine’s understanding of the vaginal microbiota’s importance in female health. We also aim to raise awareness among healthcare professionals and the general public of the significance of maintaining a healthy vaginal microbiota for better reproductive health and the prevention of gynecological diseases.

## 1. Introduction

With the development of high-throughput sequencing technology and the deepening understanding of microbial communities, more and more scholars are focusing on various microbial communities that coexist with the human body [1]. The human microbiota is a complex ecosystem consisting of trillions of microorganisms, including bacteria, viruses, fungi, and archaea, that reside in and on the human body [2]. Studies have shown that disruptions to the microbiota can have a significant impact on human health and may be associated with the development of various diseases [2,3]. Many of these studies have primarily focused on the microbiota of the gastrointestinal tract, while only a few have investigated the vaginal microbiota. There has been limited exploration of whether alterations in the vaginal microbiota could potentially impact women’s health, particularly their reproductive health [4,5,6]. Unlike the microbiota of most other parts of our body, the vaginal microbiota is highly dynamic and changes quickly. It can be influenced not only by internal factors such as a woman’s race, age, and physiological status but also by a variety of external factors such as drugs, geography, climate and environment [7,8,9,10,11,12]. With the impact of various internal and external factors, the vaginal microflora is extremely susceptible to changes and imbalances, which can have many negative effects on women’s health. According to previous studies, imbalanced microbiota is associated with a variety of reproductive disorders, including but not limited to intra-amniotic infection (IAI), spontaneous preterm birth, spontaneous abortion, premature rupture of membranes (PROM), preeclampsia, in vitro fertilization and embryo transfer (IVF-ET), bacterial vaginosis (BV), sexually transmitted diseases (STD), polycystic ovary syndrome (PCOS), gynecologic malignancies (Figure 1) [8,9,10,11,13,14,15,16,17,18,19,20]. These studies have come to our attention in recent times, and this review will mainly focus on summarizing and describing recent advances in vaginal microbiota and their role in adverse reproductive outcomes in women (Figure 2).

## 2. The Normal Vaginal Microbiota

### 2.1. Evidence Concerning Vaginal Microbiota

The vaginal microbiota coexists symbiotically with the host, and the two govern each other and influence each other [20]. On the one hand, its composition and function fluctuate according to the age of the host, sexual behavior, hormone levels and medication use [21,22,23,24]. On the other hand, this fluctuation, in turn, causes an imbalance in the vaginal microbial, allowing the overgrowth of opportunistic pathogenic bacteria and the suppression of beneficial ones, ultimately causing the host diseases, such as reproductive disorders [7,25,26]. Thus, an in-depth understanding of this interaction is essential to maintain female reproductive health. Based on the extensive application of 16S rRNA gene sequencing and metagenomic sequencing technologies, the ability to study the microbiota has greatly improved, and a better understanding of the status and role of the vaginal microbiota has been achieved [27]. The vaginal microbiota contains a wide variety of species, much more than the information obtained by traditional culture methods [4,5,6,28,29,30]. The application of sequencing technology has enriched people’s understanding of the etiology of diseases. So far, vaginal microbiota has been associated with a variety of female reproductive system diseases, especially adverse pregnancy outcomes. Some emerging pathogens may be involved in the pathogenesis of adverse pregnancy outcomes and gynecological diseases [31,32,33]. These findings are likely to lead to more effective prevention and treatment of related diseases in the future.

### 2.2. Composition and Variation of Normal Vaginal Microbiota

Female infants will acquire vaginal microbiota soon after delivery [34]. A study by Dominguez-Bello et al. based on 16S rRNA gene sequencing showed that the microbiota in the skin, nasopharynx, oral cavity, and intestine of vaginally delivered infants were similar to their mother’s vaginal microbiota, most commonly *Lactobacillus*. In contrast, those who were delivered by cesarean section at the above-mentioned sites resembled the microbiota of the mother’s skin surface and consisted mainly of *Staphylococcus*, *Corynebacterium*, and *Propionibacterium* spp. [35]. Another study found that the meconium of neonates delivered at term contained bacteria, suggesting that this intestinal microbiota was seeded before birth [36,37]. The status of the vaginal microbiota of female infants before delivery is unknown, and it is not known how different delivery methods will affect the development of the infant’s microbiota or how it will affect the subsequent health of the infant. The vaginal microbiota in childhood is mainly a mixture of skin and intestinal microorganisms, with differences in the dominant flora at different ages. As estrogen levels change in the body with age, the number of *Lactobacilli* in the vagina increases rapidly and becomes dominant [38,39]. The vagina of healthy women of reproductive age is usually dominated by *Lactobacillus*, with small amounts of *Gardnerella vaginalis*, *Corynebacterium*, *Prevotella*, *Streptococcus*, *Bifidobacterium*, *Streptococcus*, *Staphylococcus* and *Candida* [6,7,28]. In 2011, Ravel et al. classified the vaginal microbiota of women of reproductive age into five types (vaginal community structure types, CSTs) based on 16S rRNA gene sequencing technology. Four of these are dominated by *Lactobacillus* spp. (CST-I, *Lactobacillus crispatus*, CST-II, *Lactobacillus gasseri*; CST-III, *Lactobacillus inertis*, CST-V, *Lactobacillus jensenii*), and the left one is mainly composed of a variety of anaerobic bacteria (CST-IV, such as *Prevotella*, *Atopobium*, *Gardnerella*, *Megasphaera*, *Peptoniphilus*, *Sneathia*, *Eggerthella*, *Aerococcus*, *Finegoldia*, and *Mobiluncus*) [28]. It has been observed that there are variations in the types of vaginal microbiota among healthy women of different ethnic backgrounds, with *Lactobacillus* being the predominant species in the vagina of a majority of women [28]. However, the reasons for these differences are not fully understood and may be influenced by a range of factors, including hygiene practices, contraceptive practices, sexual behavior, rectal colonization, and individual genetic variations [26,40]. The vaginal microbiota of women during pregnancy tends to stabilize, with *Lactobacillus* as the dominant bacterium and low diversity. After delivery, the vaginal microbiota returns to pre-pregnancy levels [41]. As menopause is reached, estrogen levels decrease, the percentage of *Lactobacillus* in the vagina decreases, and vaginal pH increases accordingly [42,43]. In addition to the factors mentioned above, smoking, alcohol consumption, emotions, and medications can also affect the vaginal microbiota [8,9,10,44,45].

## 3. Relationship between Vaginal Microbiota and Adverse Pregnancy Outcomes

### 3.1. Vaginal Microbiota and Spontaneous Preterm Birth

Most studies are currently available on the relationship between vaginal or upper genital tract microbiota and spontaneous preterm birth [46,47,48]. Previously, based on traditional laboratory methods, pregnant women with BV were found to have a 40–84% higher risk of spontaneous preterm birth in mid to late pregnancy than healthy pregnant women, and intrauterine infection is one of the main causes of preterm birth [49,50]. Culture methods are considered the gold standard for the diagnosis of pathogenic infections; however, due to the limitations of laboratory culture techniques and available conditions, some pathogens with special requirements for growth environment or novel pathogens are often not detected by culture, which greatly affects the diagnosis and evaluation to some extent [51]. Recent studies of vaginal microbiota in pregnancy using sequencing methods not only confirmed the accuracy of previous analyses on pathogens associated with preterm birth using culture methods but also identified a new genus of bacteria associated. Han et al. used both traditional culture methods and 16S rRNA gene sequencing to analyze amniotic fluid samples from 46 patients with preterm birth and 16 asymptomatic pregnant women. Nearly two-thirds of the patients’ amniotic fluid samples with positive sequencing were found to be negative for culture, including uncultured or difficult-to-culture *Fusobacterium nucleatum*, *Bergeyella* spp., *Peptostreptococcus* spp., *Bacteroides* spp., and *Sneathia*. While 56.25% of the culture-positive specimens were found to have other pathogens after sequencing. Patients with positive gene amplification only were often accompanied by increased IL-6 levels, tissue chorioamnionitis and umbilic colitis, and early neonatal sepsis [52]. To date, a variety of bacteria genera have been found to be associated with preterm birth through culture-independent methods, including but not limited to the following taxa, e.g., *Gardnerella*, *Atopobium, Prevotella*, *Aerococcus*, *Parvimonas*, *Dialister, Sneathia*, *Megasphaera*, *Parvimonas* and *Mobiluncus* [32,33]. In addition, multiple studies have found that changes in the type and abundance of *Lactobacillus* are closely related to increases in the risk of spontaneous preterm birth. Di Giulio et al. found an association between spontaneous preterm birth and vaginal *Lactobacillus* deficiency during pregnancy after weekly sampling and analysis of cervicovaginal flora throughout pregnancy with the aid of 16S rRNA gene sequencing [53]. Shan Sun et al. also reported that women with a vaginal microbiota dominated by *Lactobacillus crispatus* had a significantly lower risk of spontaneous preterm birth than those with *Lactobacillus iners*. This association was also found to be true in both blacks and whites [16]. Their study also found that pre-pregnancy vaginal douching had a significant impact on the vaginal microbiota, which in turn affected the risk of spontaneous preterm birth. They propose that douching disrupts a healthy microbiota dominated by *Lactobacillus crispatus* and transforms it into a high-risk one [16]. This hypothesis has been supported in studies involving white individuals, but further research is needed to determine its applicability to individuals of other groups [16]. In the Hui Kan et al. study, the presence or absence of diabetes was included as an influencing factor, and it was found that the risk of spontaneous preterm birth in all women decreased with *Lactobacillus mulieris* while the risk in gestational diabetics increased with *Lactobacillus paragasseri*, *Lactobacillus gasseri*, *Streptococcus*, and *Proteobacteria* [54]. Elovitz and his team conducted a prospective cohort study of 2000 singleton pregnancies to examine the association of cervicovaginal microbiota with spontaneous preterm birth and local immunologic characteristics. Seven bacteria were found to be significantly associated with an increased risk of preterm birth, with a greater effect in African American women. Higher vaginal β-defensin two levels reduced the risk of spontaneous preterm birth associated with cervicovaginal flora in a race-dependent manner. Even in populations where the cervicovaginal flora is dominated by *lactobacilli*, low β-defensin two is still associated with an increased risk of preterm birth [55]. A recent study conducted by Chan D et al. showed that vaginal Lactobacillus depletion and high bacterial diversity causes the maternal host to increase mannose-binding lectin, IgM, IgG, C3b, C5, IL-8, IL-6 and IL-1β and to increased risk of spontaneous preterm birth [56]. Evidently, the microbiota and local immune status of the female reproductive tract during pregnancy are closely related to spontaneous preterm birth, and their intrinsic causality and the mechanisms involved deserve in-depth investigation.

### 3.2. Vaginal Microbiota and Premature Rupture of Membranes

Premature rupture of membranes (PROM), a process of spontaneous rupture of membranes before the onset of labor [57], is another adverse pregnancy closely associated with the imbalance of vaginal microbiota [58,59,60,61]. PPROM has closely related to maternal and neonatal morbidities, such as significantly lower birth weight and gestational age, early-onset neonatal sepsis, funicity, endometritis, chorioamnionitis urinary tract infection, and postpartum bacteremia [58,62]. Infection is probably one of the most important mechanisms of PROM. The study by Jayaprakash et al. looked at the microbiota in vaginal samples from 36 pregnant women who had PROM. They found that there was less *Lactobacillus vaginal* in these samples and more diversity overall. They also found that all the samples contained *Megasphaera* and *Prevotella* and that many women had *Mycoplasma hominis* and/or *Ureaplasma urealyticum*, which was associated with lower gestational age and birth weight [63]. Richard G. B et al. conducted a study of 250 pregnant women and 87 women presenting with PPROM. The results indicate that around a third of cases have vaginal dysbiosis characterized by *Lactobacillus* spp. depletion and high diversity prior to the rupture of fetal membranes and persisted following membrane rupture after comparison with pregnancies delivering at term [13]. Soon afterward, the authors recruited over 1500 early pregnancy women between 6 and 10 weeks’ gestation to investigate when vaginal microbiota shifts toward a higher diversity state in women who subsequently experienced PPROM during pregnancy. Their data demonstrate that the point about this shift emerges during the second trimester. In addition, *Prevotella*, *Streptococcus*, *Peptoniphilus* and *Dialister* may be the potentially pathogenic species associated with membrane rupture [60]. Chunmei Yan et al. similarly found an increase in pathogenic bacteria and a decrease in primary resident microbiota in the vaginal microbiota of patients with PROM [61]. They reported that *Lactobacillus iners*, *Gardnerella vaginalis*, *Prevotella bivia*, *Ochrobactrum* sp., *Prevotella timonensis*, and *Ureaplasma parvum* were increased in patients, while *Lactobacillus crispatus* and *Lactobacillus gasseri* were more abundant in the normal ones, and similarly found to have increased vaginal microbiota diversity in patients [61]. A recent study found that the presence of *Lactobacillus mulieris*, a new species of *Lactobacillus* discovered in 2020, was associated with a lower risk of PPROM. Typical pathogens such as *Megasphaera*, *Faecalibacterium* and *Bifidobacterium* were also observed in patients with PPROM [64]. The aforementioned studies in various countries based on different populations have made similar findings. Vaginal microflora may be one of the important factors in assessing the risk of premature rupture of membranes in the future.

### 3.3. Vaginal Microbiota and In Vitro Fertilization and Embryo Transfer Outcomes

In addition to being associated with a variety of natural pregnancy outcomes, the status of the vaginal microbiota also significantly affects the outcomes of in vitro fertilization and embryo transfer (IVF-ET). Hyman et al. used 16S rRNA gene sequencing to study the vaginal secretions of 30 IVF-ET patients at different times and found that vaginal species diversity varied by hormone level, and the timing of ET correlated with post-transfer outcome (live/non-live birth). Moreover, the status of the vaginal microbiota on the day of ET significantly influenced the transfer outcome, and the species diversity index could be used to determine the ET outcome [65]. A study of 91 IVF-ET cases investigated the impact of bacteria from the vagina and catheter tip on live birth rates. Hydrogen peroxide-producing *Lactobacillus* in both locations was associated with higher rates, while small amounts of potentially pathogenic bacteria were found in the vagina and cervix. *Streptococcus viridans* at the catheter tip were negatively correlated with the live birth rate. Bacterial composition at the time of ET may be crucial for successful pregnancy outcomes, with implications for reproductive medicine [66]. To investigate the effect of endometrial microbiota on reproductive outcomes in infertile patients undergoing IVF-ET, Moreno et al. performed 16S rRNA gene sequencing on endometrial biopsies collected before the week of in vitro fertilization. A non-*Lactobacillus* dominant microbiota in the endometrium was significantly associated with reduced embryo implantation rate, pregnancy/continued pregnancy, and live birth rate [67]. Another study in the same year once again suggested that abnormal vaginal microbiota was consistent with unfavorable clinical pregnancy rates in patients undergoing IVF-ET [68]. Based on gene sequencing analysis, a study found increased vaginal microbiota diversity, decreased abundance of *Lactobacillus* spp., and significant differences in vaginal microbiota composition in a population with unexplained recurrent implantation failure [18], which remotely echoes previous studies. Recent studies also showed that vaginal microbiota with *Lactobacillus*-dominant is an important favorable factor for a good outcome of frozen embryo transfer [69]. These studies indicate that vaginal microbiota dominated by *Lactobacillus* spp. is beneficial for IVF-ET outcomes. Therefore, assessment of the vaginal microbiota in this population is expected to be one of the novel initiatives to predict implantation success.

### 3.4. Vaginal Microbiota and Spontaneous Abortion

Whether vaginal infection, an outcome of the imbalanced vaginal microbiota, increases the risk of spontaneous abortion has long been debated, and current international guidelines do not recommend vaginal infection as a routine screening test [70]. However, several studies have reported an increased presence of *Mycoplasma hominis, Ureaplasma urealyticum*, *Listeria monocytogenes*, *Gardnerella vaginalis*, and other rare pathogens in the vagina of women with recurrent spontaneous abortion based on traditional serology and culture methods. To some extent, this suggests a potential link between an imbalance of the vaginal microbiota and spontaneous abortion [71,72]. A recent study based on 16S rRNA gene sequencing technology exploring the relationship between vaginal microbiota and spontaneous abortion has shown that women with early pregnancy spontaneous abortion have a high diversity of vaginal microbiota and a sharp decrease in the proportion of *Lactobacilli* [73]. However, possibly limited by current research techniques, M Al-Memar et al. had difficulty determining when the changes of the vaginal microbial and decrease in *Lactobacilli* occurred, whether it was before or sometime after pregnancy [73]. The latest study by Karen Grewal et al. also confirmed that *Lactobacillus* reduction is strongly associated with spontaneous pregnancy abortion [74]. Furthermore, they found euploid miscarriages, also called chromosomally normal miscarriages, were more strongly related to imbalanced vaginal microbiota, particularly the lack of *Lactobacillus*, compared to aneuploid miscarriage [74]. The findings of Dan Sun [75], Fenting Liu [76] et al. are consistent with the results of both mentioned above. Dan Sun et al. also found that alterations in *Lactobacillus* species similarly increased the risk of spontaneous abortion. More importantly, based on KEGG analysis, they first proposed that the decrease in *Firmicutes* in the vagina may affect the energy metabolism of the mother and, thus, lead to missed abortions [75]. This finding is new and significant, but further experiments are still needed to verify it.

Zhang et al. found a significantly higher abundance of *Atopobium*, *Prevotella*, and *Streptococcus* spp. while a lower abundance of *Lactobacillus* and *Gardnerella* spp. compared to healthy controls in a study of 10 patients with a recurrent spontaneous abortion of unknown etiology [77]. Another study on the vaginal microbiota of the recurrent spontaneous abortion population found that the diversity and composition of the vaginal microbiota of this population differed from that of the healthy population, with *Sneathia*, *Megasphaera* spp., *Pseudomonas*, *Sphingomonas*, *Rhodococcus*, *Corynebacterium*, and *Burkholderia_Caballeronia_Paraburkholderia* are strongly associated with recurrent spontaneous abortion [45]. It is worth exploring how these abnormal abundances in the vaginal microbiota in the spontaneous abortion population are involved in the development of spontaneous abortion, either individually or through interaction. In addition, unlike uncorrectable causes of spontaneous abortion, such as chromosomal abnormalities, imbalances in the vaginal microbiota can be remedied with targeted antibiotics, probiotics, and other treatments, which gives us the opportunity to prevent spontaneous abortion to some extent [45,74]. Therefore, further research is urgently needed to explore how vaginal microbiota affects the occurrence of spontaneous abortion.

### 3.5. Vaginal Microbiota and Intra-Amniotic Infection

Intra-amniotic infection (IAI) is a serious complication of pregnancy that can lead to preterm labor, fetal distress, and fetal death. It is caused by the ascending of pathogenic microorganisms from the vagina into the amniotic cavity, resulting in an inflammatory response and potentially harmful consequences for both the mother and fetus [78,79,80]. There is increasing evidence to suggest that alterations in the vaginal microbiota can contribute to the development of IAI, highlighting the importance of understanding the role of the vaginal microbiota in pregnancy [33]. Several studies have investigated the association between vaginal dysbiosis and IAI [81]. In a study by Kindinger et al., researchers analyzed the vaginal microbiota of women with PROM and found that those who developed IAI had a higher abundance of pathogenic bacteria, including *Gardnerella vaginalis* and *Prevotella bivia*, and a lower abundance of *Lactobacilli* in their vaginal microbiota compared to those who did not develop IAI [81]. Similarly, in a study by Doyle et al., researchers found that women who delivered preterm and had histological evidence of chorioamnionitis had a higher abundance of pathogenic bacteria, including *Fusobacterium nucleatum* and *Mycoplasma hominis* [82]. The exact mechanisms by which dysbiosis leads to IAI are not fully understood, but it is believed that alterations in the vaginal microbiota can lead to the overgrowth of pathogenic bacteria, which can then ascend from the vagina to the amniotic cavity and cause infection [83,84]. Dysbiosis may also lead to an inflammatory response in the vaginal and intrauterine environments, further exacerbating the risk of IAI and other obstetric complications [85]. Several strategies have been proposed for preventing IAI in women with vaginal dysbiosis. These include the use of probiotics to restore the normal vaginal microbiota, the use of antibiotics to treat bacterial infections, and the use of vaginal pH monitoring to identify women at risk for dysbiosis [86]. However, the effectiveness of these strategies in preventing IAI has not been fully established, and further research is needed to identify the optimal approach for managing dysbiosis and preventing obstetric complications.

### 3.6. Vaginal Microbiota and Preeclampsia

Preeclampsia, a devastating adverse pregnancy outcome, is one of the most serious causes of morbidity and mortality for maternal and fetal health. This condition is manifested by maternal hypertension and a multi-organ functional disturbance, such as severe proteinuria, renal insufficiency, persistent epigastric pain, hepatic dysfunction, thrombocytopenia, and pulmonary edema. The pathogenesis of preeclampsia is still unclear. Some theories suggest that a chronic inflammatory state may be involved in the development of this disease. The disturbance of vaginal microbiota may contribute to these adverse pregnancy outcomes by participating in such chronic inflammatory processes. Amarasekara et al. analyzed the bacterial composition of placental tissue from 55 primipara women with preeclampsia and 55 matched controls during cesarean section delivery. They found ten types of bacteria, including *Prevotella*, *Variovorax*, *Porphyromonas*, *Dialister*, *Anoxybacillus*, *Klebsiella pneumonia*, *Escherichia*, *Bacillus cereus*, *Salmonella*, and *Listeria* in placental tissue from seven women with preeclampsia, while none were found in the control [87]. These bacteria may be involved in the development and progression of preeclampsia. However, under what circumstances and by what route these bacteria enter the placental tissue to participate in the pathogenesis of the disease is still unknown. Lin et al. found that a higher vaginal relative abundance of *Prevotella bivia* was associated with severe preeclampsia in a study involving 88 women with severe preeclampsia and 85 controls [32]. The researchers also found that plasma levels of tumor necrosis factor (TNF) were significantly higher in the disease group than in the control group. Whether the increased relative abundance of *Prevotella bivia* is associated with increased TNF levels is unclear. Previous studies have shown that *Prevotella* in the vagina is associated with BV, pelvic inflammatory disease and a variety of adverse pregnancy outcomes. However, in a case-control study of gut microbiota in preeclampsia enrolling 26 preeclampsia women, 25 women with abnormal placental growth and 28 healthy pregnant women, *Prevotella* was found to be significantly lower in the gut microbiota of preeclampsia women compared to healthy control, which may be a protective factor for preeclampsia through regulating human immune and inflammatory response by producing short-chain fatty acid such as butyrate [88]. Current literature on this topic is limited, and further studies deserve to clarify whether the vaginal microbiota plays a role in developing preeclampsia.

## 4. Relationship between Vaginal Microbiota and Gynecological Diseases

Bacterial vaginosis (BV) is a condition that occurs due to an overgrowth of certain types of bacteria in the vagina. This overgrowth disrupts the natural balance of bacteria in the vagina and can lead to symptoms such as abnormal discharge, itching, and odor [89]. It is the most common inflammatory disease of the vagina in women of reproductive age. Both traditional culture-based microscopy and new-generation molecular techniques have shown that the vaginal manifestations of those with BV are characterized by a decrease in the proportion of the dominant vaginal bacteria, *Lactobacillus*, replaced by the predominance of pathogens, including *Gardnerella vaginalis*, *Prevotella*, *Atopobium*, *Peptostreptococcus, Mycoplasma*, *Sneathia*, and three bacterial vaginosis species (Bacterial vaginosis-associated bacteria, BVAB1, BVAB2 and BVAB3) [90,91,92]. Lev-Sagie et al. treated five patients with persistent and recurrent BV using vaginal microbiota transplants from healthy donors. After 5–21 months, four patients showed significant symptom improvement, a return to a predominant *Lactobacillus* genus in the vaginal microbiota, and no adverse effects. One patient showed partial improvement. This study confirms the role of vaginal dysbiosis in BV and demonstrates the potential of vaginal microbiota transplants as a safe and effective treatment option [92]. Furthermore, an atypical composition of vaginal microbiota can augment women’s vulnerability to STDs. Additionally, dysbiosis of the vaginal microbiota has been linked to infertility, with affected patients exhibiting elevated levels of *Chlamydia trachomatis* and *Ureaplasma urealyticum* in the reproductive tract, as well as *Gardnerella vaginalis* species in the cervix [93,94,95,96]. Recent findings have also suggested an association between the vaginal microbiota and chronic endometritis, cervical squamous intraepithelial neoplasia, and gynecological neoplasia [14,15,97,98]. As more attention is paid to vaginal and even upper genital tract microbiota, more new results on the relationship between reproductive tract microbiota and female health and reproductive disorders will emerge in the future.

## 5. Deficiencies and Perspectives of Vaginal Microbiota Studies

### 5.1. Identify Pregnant Women at High Risk and Search for Effective Treatment

The importance of identifying pregnant women at high risk of adverse perinatal outcomes due to imbalanced vaginal microbiota and determining when and how to treat them cannot be overstated. Therefore, it is crucial to identify women with imbalanced vaginal microbiota, such as those with bacterial vaginosis or other vaginal infections, and provide appropriate treatment. One of the most common methods for identifying imbalanced vaginal microbiota is through the use of molecular diagnostic techniques, such as quantitative polymerase chain reaction (qPCR) and next-generation sequencing (NGS) [99]. Once diagnosed, treatment options may include the use of antibiotics, probiotics and individualized treatment [100,101,102]. The identification and treatment of imbalanced vaginal microbiota in pregnant women are critical for ensuring positive pregnancy outcomes. Early diagnosis through molecular diagnostic techniques and appropriate treatment can prevent adverse pregnancy outcomes, such as preterm birth, low birth weight and PROM [101,103]. When and how to treat those pregnant women needs to consider both the woman’s physical condition and the safety of the fetus. Currently, treatment methods for vaginal infections in pregnant women mainly include local and oral drug therapies. Some studies have shown that local drug therapy is more effective [104,105]. However, for certain pathogens such as *Salmonella* and *Neisseria gonorrhoeae*, local drug therapy may not be suitable; thus, oral drug therapy is a better choice [106]. In order to choose appropriate drugs, it is necessary to consider the specific disease and pathogen. What’s more, antimicrobial drugs may increase the risk of pregnancy complications, such as preeclampsia, preterm birth, placental abruption, etc. [107]. Richard G. B found that erythromycin treatment of vaginal microflora disorders in patients with PROM exacerbated vaginal dysbiosis, including *Lactobacillus* depletion and increased relative abundance of *Sneathia* and increased the risk of subsequent mycotic vaginitis and neonatal sepsis [13]. The disturbance of vaginal flora caused by antibiotic therapy may aggravate the occurrence of adverse conditions in the mother and fetus [13,62,108]. Therefore, individualized treatment and probiotics should be recommended instead of antimicrobial regimens alone to treat adverse outcomes caused by vaginal dysbiosis [102].

### 5.2. Deficiencies and Perspectives of Research Techniques

The vaginal microbiota plays a crucial role in maintaining the health of women. Recent studies have highlighted the importance of understanding the composition and function of the vaginal microbiota for the diagnosis and treatment of various gynecological diseases. However, the methods currently used to study the vaginal microbiota have limitations that need to be addressed. In this part, we will discuss the shortcomings of existing methods and prospects for future research in the field of the vaginal microbiota.

Shortcomings of Current Methods: ➀ Culture-based methods: Culture-based methods are time-consuming and biased towards the detection of fast-growing microorganisms. These methods may fail to detect uncultivable or slow-growing organisms, leading to an incomplete picture of the vaginal microbiota. ➁ Polymerase chain reaction (PCR)-based methods: PCR-based methods have high sensitivity and specificity, but they are limited by their ability to detect only targeted sequences. These methods may miss the detection of a novel or uncultivable microorganisms, leading to a potential bias in the characterization of the vaginal microbiota. ➂ Next-generation sequencing (NGS) methods: NGS methods have become widely used in recent years due to their ability to provide a comprehensive and unbiased characterization of the vaginal microbiota. However, these methods can be expensive, require specialized equipment and bioinformatics expertise, and are subject to potential contamination and sequencing errors.

Prospects for Future Research: ➀ Multi-omics approaches: The integration of different omics techniques, such as metagenomics, metatranscriptomics, and metabolomics, could provide a more comprehensive understanding of the vaginal microbiota and its interactions with the host. ➁ Single-cell sequencing: Single-cell sequencing techniques could help to identify rare or uncultivable microorganisms and provide insights into the functional heterogeneity of the vaginal microbiota. ➂ Functional studies: In addition to characterizing the composition of the vaginal microbiota, functional studies could provide insights into the mechanisms underlying the interactions between the microbiota and the host and their role in health and disease.

All in all, although significant progress has been made in understanding the vaginal microbiota, there are still limitations in the methods currently used to study it. Future research using more comprehensive and unbiased methods could provide a deeper understanding of the vaginal microbiota and its role in health and disease.

### 5.3. Deficiencies and Perspectives of New Research Area

Furthermore, the current literature on vaginal or reproductive tract microbiota primarily focuses on bacterial characteristics, with limited research conducted on fungi. The majority of studies that have been reviewed rely on traditional culture methods for identifying fungal species. However, the latest high-throughput sequencing methods provide more accurate and comprehensive information on vaginal fungi compared to the limited information obtained through traditional culture methods. This advanced sequencing technology allows for a deeper understanding of the composition and diversity of vaginal fungi [29,30,109,110]. Research on the vaginal fungal flora is a rapidly growing and promising field. Examining the interactions between fungi, as well as between fungi and bacteria, from the perspective of vaginal fungi is crucial for gaining a better understanding of the vaginal microbiota. This understanding is important for preventing, treating, and managing infectious vaginosis and vaginal dysbiosis-related diseases. By studying the vaginal microbiota imbalance, we can improve the effectiveness of the prevention, treatment, and management of these diseases. It is, therefore, imperative to continue research in this area.

## 6. Conclusions

The vaginal microbiota, a critical component of the human microbiota, has been shown to play a pivotal role in a range of pregnancy outcomes, including intra-amniotic infection, spontaneous preterm birth, spontaneous abortion, premature rupture of membranes, preeclampsia, in vitro fertilization and embryo transfer and so on. Additionally, it is involved in the occurrence and progression of numerous gynecological diseases, such as bacterial vaginosis. In this review, we have examined the alterations in the vaginal microbiota across various conditions and have established the importance of vaginal microbiota for women’s health, particularly reproductive health, which significantly influences female pregnancy outcomes. Despite the existing limitations and inadequacies in current research, it is evident that with more focused attention and comprehensive investigation, we can unravel the relationship between vaginal microbiota and pregnancy outcome, as well as the underlying mechanisms involved.

## Figures and Tables

**Figure 1 microorganisms-11-00991-f001:**
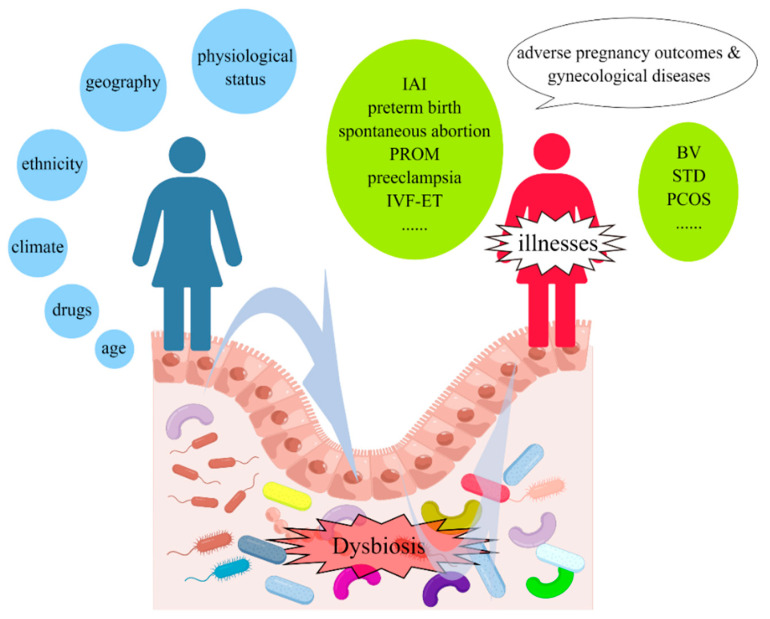
Dysbiosis of vaginal microbiota reciprocally impacts reproductive health. The composition of vaginal microbiota is affected by various factors, including but not limited to ethnicity, age, physiological status, drugs, geography, climate, and environmental factors. An imbalance of vaginal microbiota then leads to reproductive diseases, such as intra-amniotic infection (IAI), spontaneous preterm birth, spontaneous abortion, premature rupture of membranes (PROM), preeclampsia, in vitro fertilization and embryo transfer (IVF-ET), bacterial vaginosis (BV), sexually transmitted diseases (STD), polycystic ovary syndrome (PCOS), gynecologic malignancies.

**Figure 2 microorganisms-11-00991-f002:**
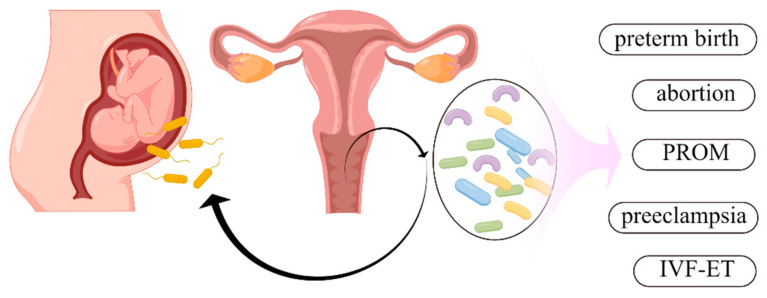
Vaginal microbiota and pregnancy outcomes. During pregnancy, vaginal microbiota is involved in intra-amniotic infection (IAI), spontaneous preterm birth, spontaneous abortion, premature rupture of membranes (PROM), preeclampsia, and in vitro fertilization and embryo transfer (IVF-ET).

## Data Availability

Not applicable.

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
