# Peer review of "Advances in Research on the Relationship between Vaginal Microbiota and Adverse Pregnancy Outcomes and Gynecological Diseases"

_microorganisms, 2023, doi:10.3390/microorganisms11040991_

Round 1

Reviewer 1 Report

The review summarizes current knowledge on associations between vaginal microbiota and pregnancy outcomes. Disturbed vaginal microbiota is considered a modifiable risk factor for adverse reproductive outcomes, therefore this topic is gaining a lot of attention of researchers and clinicians. 

These associations have been addressed in a number of recent reviews, including systematic reviews. Of the latest comprehensive reviews, the following should be mentioned: Zhu B, Tao Z, Edupuganti L, Serrano MG, Buck GA. Roles of the Microbiota of the Female Reproductive Tract in Gynecological and Reproductive Health. Microbiol Mol Biol Rev. 2022 Dec 21;86(4):e0018121. doi: 10.1128/mmbr.00181-21.

General comments

Besides three adverse pregnancy outcomes, i.e., preterm birth, premature rupture of membrane, and spontaneous abortion, the author address also in vitro fertilization and embryo transfer outcomes and some gynecological diseases. Accordingly, the research topic cannot be only defined as pregnancy outcomes, so the title should be changed.

The review does not highlight one of the most important aspect of the detrimental role of disturbed vaginal microbiota on pregnancy outcome, i.e., intra-amniotic infection, which is closely associated with the aforementioned and other pregnancy complications. It is mostly ascension of opportunistic vaginal microorganisms to the uterus that results in intra-amniotic infection, and latest advancement in the field should be outlined.

The section “Status and role of the dominant vaginal microbiota, Lactobacillus vaginalis” should be deleted or revised because 1) Lactobacillus vaginalis is not the dominant vaginal microbiota (its amount is commonly far below 1% of vaginal microbiomes) 2) lactic acid is produced by all lactobacilli, not only by L. vaginalis 3) I see no need in this section, because the idea fits that of the section “Composition and variation of normal vaginal microbiota”.

As a research gap, the authors identify lacking of comprehensive knowledge of vaginal fungal flora. This is an important research topic indeed, but in the context of adverse reproductive outcomes it is far not most acute. As disturbed vaginal microbiota is a modifiable risk factor for a number of infectious pregnancy complications, efforts primarily should be made towards identifying pregnant women at high risk of adverse perinatal outcomes and search for effective treatment strategies, i.e., what and when antimicrobial treatment should be applied.

The section Deficiencies and perspectives of research techniques in fact does not identify any gaps in knowledge in the field research techniques.

The writing style is very redundant, with repeating statements; it should be more concise and specific.

Author Response

Dear our editor and reviewers:

Hope you are well! We would like to express our sincere thanks to editors and reviewers for having this wonderful opportunity to revise our manuscript. We are very grateful to receive the valuable and constructive suggestions. We have revised our manuscript extensively according to these comments. The revisions are listed in the latter part. The page and line numbers mentioned below are in review mode.

Here, we have made a point-to-point response to the comments raised by the reviewers and resubmitted a revised manuscript on the website.

We would be glad to respond to any further questions and comments that you may have.

Looking forward to hearing from you regarding our submission.

Thank you very much.

Best regards

Sincerely yours,

Chunmei Ying, Prof, ycmzh2012@163.com.

Clinical Laboratory, Obstetrics and Gynecology Hospital of Fudan University, Shanghai, China.

The review summarizes current knowledge on associations between vaginal microbiota and pregnancy outcomes. Disturbed vaginal microbiota is considered a modifiable risk factor for adverse reproductive outcomes, therefore this topic is gaining a lot of attention of researchers and clinicians. 

These associations have been addressed in a number of recent reviews, including systematic reviews. Of the latest comprehensive reviews, the following should be mentioned: Zhu B, Tao Z, Edupuganti L, Serrano MG, Buck GA. Roles of the Microbiota of the Female Reproductive Tract in Gynecological and Reproductive Health. Microbiol Mol Biol Rev. 2022 Dec 21;86(4):e0018121. doi: 10.1128/mmbr.00181-21IF: 13.044 Q1 .

Thanks for your valuable suggestions, we have revised the article by referring to and quoting the literature you provided. Actually,the literature you recommended is very good and gives us a lot of help. Please see the revised part highlighted in yellow in the current version (Page 16, line 682-683).

General comments

Besides three adverse pregnancy outcomes, i.e., preterm birth, premature rupture of membrane, and spontaneous abortion, the author address also in vitro fertilization and embryo transfer outcomes and some gynecological diseases. Accordingly, the research topic cannot be only defined as pregnancy outcomes, so the title should be changed.

Based on your valuable comments and those of other reviewers, we reviewed the entire manuscript and changed the title to address this issue. Here's the modified version highlighted in yellow:(Page 1, line 1-36).

The review does not highlight one of the most important aspect of the detrimental role of disturbed vaginal microbiota on pregnancy outcome, i.e., intra-amniotic infection, which is closely associated with the aforementioned and other pregnancy complications. It is mostly ascension of opportunistic vaginal microorganisms to the uterus that results in intra-amniotic infection, and latest advancement in the field should be outlined.

Your suggestion is excellent and we have added this part. At the same time, we added the preeclampsia section. Thank you again for your valuable comments, so that our manuscript richer and better. Here's the modified version highlighted in yellow:(Page 10-11, line 388-445).

The section “Status and role of the dominant vaginal microbiota, Lactobacillus vaginalis” should be deleted or revised because 1) Lactobacillus vaginalis is not the dominant vaginal microbiota (its amount is commonly far below 1% of vaginal microbiomes) 2) lactic acid is produced by all lactobacilli, not only by L. vaginalis 3) I see no need in this section, because the idea fits that of the section “Composition and variation of normal vaginal microbiota”.

Thank you for your advice. We feel enlightened by your guidance. Therefore, we have removed this section. Here's the modified version highlighted in yellow:(Page 9, line 174-189-164).

As a research gap, the authors identify lacking of comprehensive knowledge of vaginal fungal flora. This is an important research topic indeed, but in the context of adverse reproductive outcomes it is far not most acute. As disturbed vaginal microbiota is a modifiable risk factor for a number of infectious pregnancy complications, efforts primarily should be made towards identifying pregnant women at high risk of adverse perinatal outcomes and search for effective treatment strategies, i.e., what and when antimicrobial treatment should be applied.

Thanks for your suggestion, we have modified and improved the whole part. Your suggestion is really constructive. Here's the modified version highlighted in yellow:(Page 12-14, line 489-589).

The section Deficiencies and perspectives of research techniques in fact does not identify any gaps in knowledge in the field research techniques.

Thanks for your suggestion, we have modified this part. (Page 12-13, line 518-560).

The writing style is very redundant, with repeating statements; it should be more concise and specific.

Thanks for your comments, we have revised the whole article and deleted the miscellaneous parts. You can see our changes and improvements in the modified version.

Reviewer 2 Report

The paper is well written and well referenced.

- The title and abstract have different focuses. If the focus of the article is intended to be on the vaginal microbiome and pregnancy outcomes then make the abstract have that focus. 

-Line 26. "Recent studies have found" is personification. 

-Line 109 This sentence reads somewhat racially insensitive.  Please modify it to soften the implications that people of color are less clean and more sexually active. 

-Line 165. Please soften this language to be less racially insensitive. "  How- ever, this hypothesis holds true for whites but not for blacks[14], and it remains to be explored in further studies." 

-2.1. Research status of vaginal microbiota. This header needs to be changed to something like evidence concerning vaginal microbiome.   

Section 4 on gyn does not seem to fit with the rest of the paper.

-Suggested edit for Figure 1 caption. Vaginal microbiota and pregnancy outcomes.

-Suggested edit for Figure 2 caption. Figure 2. Interactions between vaginal microbial and pregnancy outcomes  The figure itself is  confusing.  You show what looks like a non pregnant woman standing and then what looks like the same woman, possibly now pregnant sitting with an IV.  The images don't make sense.  What are you trying to show in the figure? 

Author Response

Dear our editor and reviewers:

Hope you are well! We would like to express our sincere thanks to editors and reviewers for having this wonderful opportunity to revise our manuscript. We are very grateful to receive the valuable and constructive suggestions. We have revised our manuscript extensively according to these comments. The revisions are listed in the latter part. The page and line numbers mentioned below are in review mode.

Here, we have made a point-to-point response to the comments raised by the reviewers and resubmitted a revised manuscript on the website.

We would be glad to respond to any further questions and comments that you may have.

Looking forward to hearing from you regarding our submission.

Thank you very much.

Best regards

Sincerely yours,

Chunmei Ying, Prof, ycmzh2012@163.com.

Clinical Laboratory, Obstetrics and Gynecology Hospital of Fudan University, Shanghai, China.

The paper is well written and well referenced.

- The title and abstract have different focuses. If the focus of the article is intended to be on the vaginal microbiome and pregnancy outcomes then make the abstract have that focus. 

Thank you for the feedback. I apologize for any confusion caused by the discrepancy between the title and abstract. We have updated the abstract to better reflect the focus of the article on the relationship between the vaginal microbiota and pregnancy outcomes and gynecological diseases in women. Thank you for bringing this to our attention. Please see the revised part highlighted in yellow in the current version (line 1-36, page 1). 

-Line 26. "Recent studies have found" is personification. 

I apologize for my confusion. I made a mistake in my previous sentence. Here's the modified version highlighted in yellow:(Page 1-2, line 45-48).

-Line 109 This sentence reads somewhat racially insensitive.  Please modify it to soften the implications that people of color are less clean and more sexually active. 

I apologize for any offense caused by the original sentence. Here's a modified version that removes any implications about people of color being less clean or more sexually active. (Page 5-6, line 162-165).

-Line 165. Please soften this language to be less racially insensitive. "  How- ever, this hypothesis holds true for whites but not for blacks[14], and it remains to be explored in further studies." 

Thank you for bringing this to my attention. I apologize for any offense caused by my previous wording. A more appropriate way to phrase this sentence could be in the modified version highlighted in yellow. (Page7, line 229-230).

-2.1. Research status of vaginal microbiota. This header needs to be changed to something like evidence concerning vaginal microbiome.   

Thank you for your feedback. I agree that "evidence concerning vaginal microbiome" would be a more accurate and descriptive header for this topic. Here's the modified version highlighted in yellow (Page 4, line 98).

Section 4 on gyn does not seem to fit with the rest of the paper.

Based on your comments and those of other reviewers, we reviewed the entire manuscript and changed the title to address this issue.

-Suggested edit for Figure 1 caption. Vaginal microbiota and pregnancy outcomes.

Sorry, we got the order of Figures 1 and 2 wrong the first time we submitted. We contacted the editor to make the change and got approval. But you don't seem to see the changed order. This is Figure 2. Thank you for your suggestion. We have made the necessary changes to the title and figure. Here's the modified version highlighted in yellow (Page 4, line 91-95).

-Suggested edit for Figure 2 caption. Figure 2. Interactions between vaginal microbial and pregnancy outcomes  The figure itself is  confusing.  You show what looks like a non pregnant woman standing and then what looks like the same woman, possibly now pregnant sitting with an IV.  The images don't make sense.  What are you trying to show in the figure? 

Thank you for your advice. In the meantime, I apologize for any confusion. We have modified the title, note and content of this picture according to your valuable comments. This figure should to be Figure 1 and the title now is Dysbiosis of vaginal microbiota reciprocally impacts reproductive health. And what we want to show is that :The composition of vaginal microbiota is affected by various factors, including but not limited to ethnicity, age, physiological status, drugs, geography, climate, and environmental factors. Imbalance of vaginal microbiota then leads to reproductive diseases, such as intra-amniotic infection (IAI), spontaneous preterm birth, spontaneous abortion, premature rupture of membranes (PROM), preeclampsia, in vitro fertilization and embryo transfer (IVF-ET), bacterial vaginosis (BV), sexually transmitted diseases (STD), polycystic ovary syndrome (PCOS), gynecologic malignancies. Hope you can be satisfied with our modification and improvement. Here's the modified version highlighted in yellow (Page 3-4, line 78-86).

Round 2

Reviewer 1 Report

The authors have largely revised the paper and mostly addressed the issues I pointed out